# Stem Cell-Based Therapies for Inflammatory Bowel Disease

**DOI:** 10.3390/ijms23158494

**Published:** 2022-07-31

**Authors:** Hua-Min Zhang, Shuo Yuan, Huan Meng, Xiao-Ting Hou, Jiao Li, Jia-Chen Xue, You Li, Qi Wang, Ji-Xing Nan, Xue-Jun Jin, Qing-Gao Zhang

**Affiliations:** 1Key Laboratory of Natural Resources of Changbai Mountain & Functional Molecules, Ministry of Education, Molecular Medicine Research Center, College of Pharmacy, Yanbian University, Yanji 133002, China; zhm4481@163.com (H.-M.Z.); yuanqiqi0710@yeah.net (S.Y.); jxnan@ybu.edu.cn (J.-X.N.); 2Chronic Disease Research Center, Medical College, Dalian University, Dalian 116622, China; 15563776072@163.com (H.M.); h52110609@163.com (X.-T.H.); lijiao232343@163.com (J.L.); 18640957697@163.com (J.-C.X.); ly18742591081@163.com (Y.L.); wwwangqi163@163.com (Q.W.); 3Department of Immunology and Pathogenic Biology, College of Basic Medicine, Yanbian University, Yanji 133002, China

**Keywords:** inflammatory bowel disease, Crohn’s disease, ulcerative colitis, stem cell therapy, mesenchymal stem cells, hematopoietic stem cells

## Abstract

Inflammatory bowel disease (IBD) is a chronic, relapsing disease that severely affects patients’ quality of life. The exact cause of IBD is uncertain, but current studies suggest that abnormal activation of the immune system, genetic susceptibility, and altered intestinal flora due to mucosal barrier defects may play an essential role in the pathogenesis of IBD. Unfortunately, IBD is currently difficult to be wholly cured. Thus, more treatment options are needed for different patients. Stem cell therapy, mainly including hematopoietic stem cell therapy and mesenchymal stem cell therapy, has shown the potential to improve the clinical disease activity of patients when conventional treatments are not effective. Stem cell therapy, an emerging therapy for IBD, can alleviate mucosal inflammation through mechanisms such as immunomodulation and colonization repair. Clinical studies have confirmed the effectiveness of stem cell transplantation in refractory IBD and the ability to maintain long-term remission in some patients. However, stem cell therapy is still in the research stage, and its safety and long-term efficacy remain to be further evaluated. This article reviews the upcoming stem cell transplantation methods for clinical application and the results of ongoing clinical trials to provide ideas for the clinical use of stem cell transplantation as a potential treatment for IBD.

## 1. Introduction

Inflammatory bowel disease (IBD), including both ulcerative colitis (UC) and Crohn’s disease (CD), is a chronic nonspecific inflammatory disease of the intestine involving the ileum, colon, and rectum [1]. The incidence of IBD varies somewhat from region to region, with IBD incidence being higher in women than in men in Europe and the United States. In contrast, the situation in Asian countries is the opposite, and the overall incidence still shows an increasing trend [2,3]. The pathogenesis of IBD is still unclear and is mainly thought to be related to genetics, infection, immunity, and intestinal microecology, but there is a lack of uniformity [4].

Stem cells can differentiate into different degrees and types of progenies, which constitute the basic tissue structure of the human body. They are characterized by their ability to increase indefinitely with various functions of progeny cells and their ability self-renew. Stem cells can be classified into three main types according to their different roles and origins: embryonic stem cells, induced multipotential stem cells, and adult stem cells [5]. In recent years, there have been breakthroughs in the research of stem cells in the medical field, and they have been widely used in the field of IBD. With the emergence of stem cell therapy medical technology, stem cell therapy for IBD has received much attention from the industry. At present, the stem cells used for IBD treatment include hematopoietic stem cells (HSC), mesenchymal stem cells (MSC), and intestinal epithelial stem cells. Among them, intestinal epithelial stem cells are the best source.

Nevertheless, because the inception of intestinal epithelial stem cells is limited and cannot be expanded in culture for a long time in vitro, the current research primarily applies hematopoietic stem cell transplantation (HSCT) and MSC therapy. The idea of stem cell transplantation for IBD first came from patients with combined hematological and malignant tumors. In 1993, Drakos et al. reported the first case of CD in remission after HSCT [6]. Kashyap et al. then observed one patient with CD in combination with non-Hodgkin’s lymphoma who achieved a 7-year clinical remission after autologous HSCT [7]. Subsequently, autologous and allogeneic stem cell transplantation gradually began to shift from opportunistic applications to targeted use for the treatment of patients with refractory and combined multiple comorbidities of IBD. In 2005, Oyama et al. reported 12 cases of refractory CD, 11 of which achieved clinical remission with autologous HSCT [8]. Reports such as these are increasing year by year in recent years. In this paper, a search of the published literature was conducted by exploring by the library databases such as PubMed and Web of Science. The following search terms related to key issues were set up for the search, including “inflammatory bowel disease”, “Crohn’s disease”, “ulcerative colitis”, “pathogenesis”, “intestinal mucosal barrier”, “oxidative stress”, “angiogenesis”, “therapy”, “stem cells”, “stem cell therapy”, “hematopoietic stem cells”, and “mesenchymal stem cells”. Here, we reviewed the research progress of pathogenesis, pharmacological treatments, and emerging stem cell-based approaches in IBD, to provide targeted suggestions and new therapeutic strategies for the clinical treatment of IBD patients.

## 2. The Pathogenesis of Inflammatory Bowel Disease

### 2.1. Genetic Factors

Numerous genome-wide association studies have shown that multiple genetic diversities cause IBD. Over 200 genetic loci have been identified that are associated with IBD as well as 70% of them are also present in other complex autoimmune and immunodeficiency diseases [9]. Genome-wide association scan data revealed 201 loci currently identified as associated with IBD [10], including 41 CD-specific loci and 30 UC-specific loci, which may result in clinical, endoscopic, and differences in phenotypic characteristics on histology [11]. Loci associated with susceptibility for IBD include those coding for barrier function, epithelial repair, microbial defense, and other biological behaviors, which have essential roles in the pathogenesis of IBD [12]. Genetic variants that increase CD susceptibility are mainly associated with innate immunity, autophagy, and phagocytosis, while genetic variants related to UC susceptibility are primarily related to intestinal barrier function [13].

### 2.2. Environmental Factors and Lifestyle

Environmental factors and lifestyle play an important role in promoting the disease in genetically susceptible individuals with IBD. Initially, smoking was an environmental risk factor associated with IBD [14]. Smoking aggravates the patient’s condition and increases the probability of surgical procedures. In contrast, smoking cessation rapidly reduces the incidence of extraintestinal manifestations in IBD patients and reduces hormone doses [15]. Nicotine in tobacco affects nicotinic acetylcholine receptors distributed in intestinal epithelial cells, which alters cytokine levels such as interleukin-8 (IL-8) and tumor necrosis factor-α (TNF-α), thereby increasing microvascular thrombosis. Furthermore, the development of CD due to smoking may be related to the number of smoking-related single nucleotide polymorphisms (SNPs) in patients. Smoking-related SNPs are positively associated with the risk of surgical procedures [16]. In addition, a high-fat and high-cholesterol diet predisposes to an increased risk of IBD. Dietary levels of linoleic acid and n-3 polyunsaturated fatty acids are significantly and negatively associated with IBD development [17]. A diet of saturated fatty acids, increases the risk of intestinal inflammation in mice [18]. Moreover, vitamin D is involved in the regulation of intestinal immune function and plays an essential role in the pathogenesis of IBD and the active phase of the disease. The presence of vitamin D deficiency in IBD patients (especially in CD patients) and serum vitamin D levels are negatively associated with the risk of developing IBD [19].

### 2.3. Intestinal Mucosal Barrier

The intestinal mucosal barrier consists of mechanical, chemical, and immune barriers in conjunction with biological barriers (Figure 1). Among them, damage to any of the barriers can disrupt the intestinal mucosal barrier, resulting in impaired function and inducing enteric-derived infections that lead to IBD [20].

#### 2.3.1. Mechanical Barrier

The structural basis of the mechanical barrier is the intact intestinal epithelial cells (IECs) and the apical junction complex (AJC), which mainly include tight junctions (TJ) and adhesive junctions (AJ) between epithelial cells. It controls the adhesion and barrier function between epithelial cells and regulates intracellular signaling pathways and transcription [21]. Tight junctions are a physical barrier, which can regulate the permeability of intestinal mucosa and prevent antigenic substances from entering the intestinal mucosa, which can induce IBD [22]. This is consistent with the increased gut TJ disruption observed in IBD patients as well as the barrier defect after the disruption. It is worth mentioning that the TJ is mainly composed of zonula occludens (ZO) proteins, transmembrane proteins (junctional adhesion molecules, occludins, and claudins), and cytoskeletal structures [23]. In conclusion, the integrity of the intestinal mechanical barrier function can resist the invasion of IBD, and the inflammatory response of IBD can further damage the mechanical barrier function. 

#### 2.3.2. Chemical Barrier

The intestinal mucosal chemical barrier is less studied, but it has a relatively important role in the body. The loose mucus layer covering the intestinal epithelial cells and the mucins in the mucus together constitute the intestinal mucosal chemical barrier. The intestinal mucus layer plays a major role in intestinal protection against mechanical, chemical, and biological attacks and helps maintain the balance of the intestinal environment [24]. The mucus is a viscous fluid secreted by cupular cells, large reticulated polymers rich in mucins and glycoproteins [25]. Mucin (MUC) 2 is the main component of the intestinal mucus and can prevent the invasion of pathogenic bacteria and their adhesion to the intestine.

#### 2.3.3. Microbial Barrier

The microbial barrier is the normal commensal flora in the gut that is resistant to colonization by foreign strains. The gut symbiotic flora and the host form an interdependent and interactive micro-ecosystem in the micro spatial structure, which constitutes microbial homeostasis and is an important protective barrier against pathogens. Inflammatory bowel disease is associated with homeostasis imbalance, mainly manifested by a reduction in gut microbial diversity [26]. The balance between commensal and potentially pathogenic microbes is altered, such as increased invasive flora and decreased protective bacteria. When microbial homeostasis is disrupted, intestinal colonization resistance is greatly reduced, which can lead to the colonization and invasion of opportunistic pathogens in the intestine, thereby increasing the risk of eliciting host immune responses and inducing or even promoting the development of IBD [27]. Inflammatory bowel disease can cause abnormal metabolic regulation in vivo, and the resident intestinal flora can suppress the expression of pro-inflammatory cytokine genes by secreting short-chain fatty acids (SCFAs), vitamins, and other beneficial active metabolites [28]. The imbalance of the biological barrier of microbial homeostasis, whether as a cause or a consequence, plays an important role in the occurrence and development of IBD. To sum up, various reasons, including the destruction of the biological barrier caused by IBD itself, will lead to the increase of pathogenic bacteria and the excessive activation of abnormal immune responses, thereby triggering IBD, resulting in a vicious circle.

#### 2.3.4. Immune Barrier

Inflammatory bowel disease is associated with a disruption of the immune response, and there are differences in immune dysregulation among different types and degrees of IBD. The gut-associated lymphoid tissue (GALT), secretory antibodies, and mesenteric lymph nodes constitute the intestinal mucosal immune barrier, responding to antitoxins, antigens, and potentially harmful organisms [29]. The GALT produces immunoglobulin A (IgA), forming IgA antigen complexes with antigenic material. After binding to receptors on M cells, the antigen is transferred to the lamina propria and then presented to the essential specialized antigen-presenting cells in the body, dendritic cells (DCs). In the mice model of IBD, DCs induce differentiation of initial T-cells to Th1 cells by secreting IL-12 and producing large amounts of interferon (IFN)-γ to mediate the intestinal mucosal inflammatory response [30,31].

As an essential component of the intrinsic immune system and the first line of defense for pathogens invading the human immune system, neutrophil infiltration is closely related to the development and progression of ulcerative colitis [32]. On the one hand, neutrophils can clear pathogens by promoting inflammation through direct phagocytosis or by releasing intracellular substances neutrophil extracellular traps (NETs), promoting mucosal healing and inflammation regression [33]. On the other hand, the NETs released by neutrophils induce apoptosis of epithelial cells, disrupting the integrity of tight junctions and adherent junctions and their impairment of intestinal epithelial barrier function in the pathogenesis of mucosal inflammation in acute colitis [34,35]. Meanwhile, reactive oxygen species (ROS) produced by neutrophils are the primary mediators of phagocytosis and killing, but their production also damages lipids and proteins, altering their function [36].

Macrophages will initiate various cell polarization pathways upon stimulation by different cytokines, chemokines, and signaling molecules. Macrophages were first found to initiate all immune responses, including T-cell adaptive immune response, B-cell adaptive immune response, polarization-related T-helper 1 (Th1) cell, and Th2 cells, thus dividing macrophages into M1-type macrophages and M2-type macrophages [37]. The M1 macrophages are predominantly expressed at the onset of inflammation and secrete large amounts of pro-inflammatory factors that promote an increased inflammatory response. In contrast, to counteract the excessive inflammatory response at a later stage, macrophages can be converted from M1 to M2 phenotype to suppress inflammation protecting the host from excessive damage and promoting wound healing. Therefore, regulation of macrophage polarization may be effective in alleviating IBD [38].

T-cells in the intestinal mucosa of CD patients are mainly Th1 cells, while T-cells in the intestinal mucosa of UC patients are mainly Th2 cells that can secrete transforming growth factor-β (TGF-β) and IL-5 [39]. In addition, Th17 cells, which can secrete large amounts of cytokines such as IL-17A, IL-17F, IL-21, and IL-22, are also thought to influence the development of IBD [40]. Regulatory T-cells (Tregs) can maintain homeostasis in the intestine, so targeting Treg cells can also be an alternative strategy to control excessive inflammation in IBD [41].

Furthermore, natural killer (NK) cells are important immune cells of the body, not only associated with antitumor, antiviral infection, and immune regulation but also involved in hypersensitivity reactions and autoimmune diseases in some cases. The specificity of NK cell surface markers is relative compared to T-cells and B-cells. Activated NK cells can synthesize and secrete a variety of cytokines that exert immunomodulatory and hematopoietic effects as well as the direct killing of target cells [42].

In addition, the immunotherapies currently under development for IBD also focus on the downstream pathways of different cytokine-related pathways, such as Nucleotide-binding oligomerization domain 2 (NOD2), Toll-like receptors (TLRs), and Signal Transducer and Activator of Transcription (STAT), are blocked and inhibited, thereby reducing inflammation in IBD.

Ramanan et al. found that NOD2 gene expression deficient mice had insufficient intestinal cup cells and an increased number of pro-inflammatory bacteria mimics, leading to an increased CD incidence [43]. Defective autophagy in CD patients with NOD2 mutations and bacterial lipopolysaccharide recognizes damage-associated molecular patterns (DAMPs), activates TLRs, promoting cytokine-mediated inflammatory responses in the intestine such as TNF-α. The NOD2 mutations are associated with injuries such as CD fibrous stenosis, ileal lesions, and intestinal penetrating lesions [44]. It should be noted that NOD2 gene mutations in CD patients are more common in European and American countries but not reported in Asia, so there may be regional and ethnic differences in NOD2 gene mutations. In addition, miR-146a controls NOD2 expression and may be involved in the pathogenesis of IBD by regulating NOD2 downstream signaling pathways [45].

Toll-like receptors are essential pattern recognition receptors that link innate and acquired immunity, and dysfunctional TLRs can mediate and maintain chronic inflammatory responses, which are closely associated with the pathogenesis of IBD. Studies have shown that TLR2 mRNA expression in intestinal mucosal tissue is significantly higher in IBD patients compared to normal subjects, while TLR1 R80T and TLR2 R735G heterozygous carriers are at increased risk of total colitis [46]. The TLR4 was overexpressed in intestinal epithelial cells of IBD patients. In CD patients, TLR4 is expressed chiefly on the luminal side of intestinal mucosal epithelial cells, whereas in UC patients, TLR4 is mainly expressed on the basal side of mucosal epithelial cells. In addition, TLR4 D299G has a synergistic effect with NOD2 gene mutation, leading to an earlier age of CD onset [47]. The expression level of TLR5 is associated with UC onset, while TLR3 and TLR9 have a protective effect on the intestinal mucosa and inhibit IBD onset [48].

The STAT family is a class of intracellular proteins that both signal and activate transcriptional functions and contains seven members (STAT1-4, 5A, 5B, and 6), of which STAT3 is most closely associated with immunosuppression and is the only member of the family whose genetic defect leads to embryonic necrosis [49]. A specific interaction between nuclear factor kappa-B (NF-κB) and STAT3 pathways can jointly promote inflammatory responses [50]. At the same time, the occurrence of IBD is associated with the dysregulation and sustained activation of the immune response in the intestine.

### 2.4. Oxidative Stress

Typically, the production of ROS in the body is in balance with antioxidant protection mechanisms. The ROS levels are elevated when exposed to UV radiation, smoking, alcohol consumption, non-steroidal anti-inflammatory drugs, infections, ischemia-reperfusion injury, and multiple inflammatory responses [51]. Intracellular ROS are produced in various ways, varying from tissue to tissue, the most prominent of which is a reduced coenzyme II (nicotinamide adenine dinucleotide phosphate (NADPH) oxidase (NOX) complex. This complex is a transmembrane protein widely present in the cell membrane, mitochondria, peroxisomes, and endoplasmic reticulum [52,53]. In addition, a portion of ROS is also produced in the electron transport chain of the inner mitochondrial membrane. Although most oxygen is ultimately reduced to water by cytochrome oxidase in the mitochondria, some oxygen is still incompletely reduced, and ROS are generated [54]. The gastrointestinal tract is a crucial source of ROS. Although the intestinal epithelium has a protective barrier, certain substances and pathogens that are ingested can stimulate inflammatory responses by secretion of inflammatory factors and other inflammatory mediators by intestinal epithelial cells, polymorphonuclear neutrophils, and macrophages, which can promote the development of oxidative stress [51]. Excessive ROS and reactive nitrogen species (RNS) in intestinal tissues cause lipid peroxidation, DNA damage, and apoptosis, which also lead to the impairment of enzymatic and non-enzymatic antioxidant mechanisms such as superoxide dismutase (SOD), reduced glutathione (GSH), catalase (CAT), and malondialdehyde (MDA) mechanisms, ultimately causing colonic damage [55]. Activated inflammatory cells can stimulate the NADPH oxidase system and inducible nitric oxide synthase (iNOS) to produce large amounts of superoxide and nitric oxide, respectively, as well as to release large amounts of myeloperoxidase (MPO), inducing colonic inflammation and pathological changes [56].

Nuclear factor-erythroid 2-related factor-2 (Nrf2) is a primary transcriptional regulator of the cellular antioxidant stress program, a member of the CNC family that coordinates the activation of cytoprotective genes to defend against exogenous stress and oxidative stress [57]. The activation of Nrf2 significantly inhibited ROS production, attenuated the inflammatory response, promoted cell survival, and improved intestinal epithelial cells’ redox and inflammatory status (Figure 2) [58]. By activating the expression of Nrf2 and its downstream multiple antioxidant enzymes, it can effectively reduce intestinal epithelial damage and apoptosis in a porcine jejunal epithelial cell oxidative stress model and improve intestinal mucosal barrier permeability, confirming that Nrf2 can inhibit oxidative stress and reduce intestinal inflammation [59]. In addition, Nrf2 can cross-regulate intestinal oxidative stress with autophagy. The p62 protein, located at the autophagosome formation site, binds to the autophagy localization protein LC3 and ubiquitinated proteins to participate in the autophagic process, and shearing the Keap1 interaction domain of p62 protein pre-mRNA significantly increases Keap1 content, enhances Nrf2 ubiquitination, and inhibits its target gene expression [60]. Notably, Keap1 gene ablation can contribute to the accumulation of ubiquitin aggregates and impaired autophagy activation, suggesting that Keap1-p62 conjugates are also involved in the degradation of ubiquitinated proteins [61]. Thus, various intracellular autophagic networks and anti-oxidative stress processes can be regulated by Keap1-p62 interactions. Harada et al. showed that defective autophagy activates the Nrf2/Keap1/Heme Oxygenase-1 (HO-1) pathway and attenuates indomethacin-induced intestinal epithelial damage in mice [62]. This suggests that the above two pathways can be coordinated to maintain normal intestinal barrier function under pathophysiological conditions.

### 2.5. Angiogenesis

Angiogenesis is the process of forming new blood vessels by outgrowth or other means based on existing micro-vessels. Usually, the growth and inhibition of blood vessels in the body are in a dynamic balance to maintain a relatively stable state, and it can lead to many diseases once imbalanced. In the early stages of IBD, serum levels of pro-angiogenic factors are elevated and imbalanced with the action of angiogenic inhibitors, thus leading to pathological angiogenesis. Scaldaferri et al. extracted intestinal samples from patients with IBD and controls and measured the expression levels of vascular endothelial growth factor-A (VEGF-A) and its receptors VEGFR-1 and VEGFR-2 well as the induction of VEGF-A in human intestinal microvascular endothelial cells angiogenesis. The results showed that both VEGF-A and VEGFR-2 levels were elevated in IBD patients compared to controls [63]. It was found that VEGF-A was elevated in a mouse model of DSS-induced colitis. The overexpression of VEGF-A increased angiogenesis in the intestinal mucosa and stimulated leukocyte adhesion, worsening the disease in mice, suggesting that IBD is accompanied by angiogenesis in chronic inflammation.

The immune-inflammatory response and angiogenesis in IBD are mutually reinforcing processes. During the inflammatory process, a series of cellular signals first increases the expression of vascular endothelial cell adhesion molecules, which further increases the recruitment of leukocytes. The interaction of the recruited leukocytes with the vascular endothelium leads to the rolling of leukocytes along the vascular endothelium, allowing them to adhere firmly to the vascular endothelium, and the adherent leukocytes migrate across the vascular endothelial cell layer into the surrounding tissues in a transmembrane fashion. Once they reach the tissues, these immune cells release a variety of pro-inflammatory and angiogenic mediators, including cells (e.g., leukocytes, platelets) and biochemical molecules (e.g., cytokines or chemokines) [64], which can potently promote angiogenesis through different mechanisms. In summary, the inflammatory response can promote pathological angiogenesis.

Many factors in inflammatory tissues can regulate angiogenesis (Figure 3). Hypoxia is a necessary angiogenic stimulus, as vascular smooth muscle cells proliferate and produce hypoxia-inducible factor-1α (HIF-1α) in large quantities under hypoxic conditions. HIF-1α is an oxygen regulatory protein tightly regulated by oxygen partial pressure, and HIF-1α acts on VEGF target genes. The expression levels of VEGFR mRNA increase with higher levels of HIF-1α expression, showing a significant positive correlation [65]. The VEGF is the most direct vascular endothelial cell pro-divider, and VEGF binds to receptors and exerts a series of biological functions to promote angiogenesis. In addition, inflammatory lesions contain many inflammatory cells (such as macrophages, mast cells, lymphocytes, and fibroblasts) that directly or indirectly release many pro-angiogenic factors. These factors include the VEGF family and its receptors, integrins αvβ3 and αvβ5, monocyte chemotactic protein (MCP), platelet-derived growth factor-BB (PDGF-BB), matrix metalloproteinase (MMP), and TGF-β1, all of which can promote angiogenesis. Platelets are significantly increased in patients with IBD. It has been shown that angiogenesis-promoting factors and inhibitory factors are stored in different platelet α granules, which participate in angiogenesis by selectively releasing different types of angiogenic regulatory factors from platelet α granules, as well as actively mobilizing various membrane surface receptors to participate in their adhesion and aggregation, contributing to angiogenesis [66]. In addition, platelet α granules secrete proteases that degrade the basement membrane and act as inhibitors of angiogenic factors, thereby releasing bound angiogenic factors to aid in angiogenesis. Rutella et al. found that the expression of intercellular adhesion molecule-1 (ICAM-1), chemokines, and integrin αvβ3 increased in HIMEC under TNF-α stimulation, which promoted platelet adhesion and the release of pro-angiogenic factors VEGF and CD40L from platelets. Thus, platelet increase was closely related to angiogenesis [67]. The vascular endothelium changes during the initial inflammation phase are functional and include vasodilation, increased endothelial cell permeability, and blood cell spillage. The increased blood flow in the dilated vessels and the resulting shear stress can also promote angiogenesis [68]. In addition, bacterial flora plays a vital role in VEGF production and angiogenesis. Cane et al. showed that wild-type C1845 bacteria promote VEGF production by human intestinal T84 cells, and Afa/Dr *Escherichia coli* C1845 bacteria mediate the activation of F1845 adhesin-dependent-receptor decay acceleration factor (DAF), through the Src protein kinase-dependent mechanism pathway, activating serine-threonine kinase (Akt) and extracellular signal-regulated kinase (ERK), which induce gene transcription leading to elevated VEGF expression, thereby promoting angiogenesis [69]. Furthermore, vascular endothelial cells can also regulate immune cell function through contact with immune cells, thereby regulating the body’s intrinsic and acquired immunity.

## 3. Pharmacological Treatments of Inflammatory Bowel Disease

With advances in pharmaceutical research and development, more and more medicines for the treatment of IBD have been developed, including biologics and small chemical molecules, which has led to increasing diversity in the choice of medications used to treat IBD, with the primary goal of inducing remission and maintaining remission (Table 1). However, individualized treatment protocols are mainly adopted in clinical practice.

### 3.1. Small Molecule Medications

#### 3.1.1. Amino Salicylate

Currently, oral 5-aminosalicylic acid (5-ASA) preparations are commonly used clinically to treat mild to moderately active ulcerative colitis. These drugs were first used in the treatment of rheumatoid arthritis, and in the process of clinical application, they were found to be able to relieve the symptoms of patients with IBD, thus starting to be used in the treatment of IBD. However, as 5-ASA can cause gastrointestinal side effects such as nausea and vomiting, new 5-ASA formulations such as suppositories and enemas were developed. Early studies have shown that 5-ASA is similar to salazosulfapyridine (SASP) in that it is antibacterial, inhibits prostaglandin and NF-κB activity, and scavenges reactive oxygen metabolites. Nevertheless, there is recent evidence that these effects are ineffective and are not recommended for use in conventional therapy [84].

#### 3.1.2. Glucocorticoids

Glucocorticoids are often used as a supplement when 5-ASA drugs are not effective, and topical hormone therapy (including prednisone and hydrocortisone) can be effective in relieving moderate to severe UC and CD. Clinicians have used glucocorticoids for more than half a century, and in 1954 Truelove and Witts found that glucocorticoids had a significant therapeutic effect on UC. Glucocorticoids can exert their ability to modulate the immune response by interacting with glucocorticoid receptors in the cell nucleus, while these receptors can inhibit the expression of adhesion molecules and the transport of inflammatory cells to the intestine [71]. However, glucocorticoids have been found to have specific side effects on patients during the clinical application, including diabetes, osteoporosis, systemic edema, and infections, which lead to the use of glucocorticoids as long-term therapeutic agents, although they can alleviate the symptoms of clinical IBD patients. Studies have also found that geriatric patients have a high drug dependence on glucocorticoids, which is an important factor limiting the long-term use for the treatment of IBD [72,73].

#### 3.1.3. Immunomodulators

When patients cannot tolerate glucocorticoids, azathioprine, and its active metabolite 6-mercaptopurine, methotrexate and cyclosporine can be further applied to treat UC and CD [85]. Methotrexate exerts its therapeutic effects in patients by inhibiting several enzymes involved in the folate metabolic pathway. High doses of methotrexate can kill cancer cells or inhibit their proliferation by inhibiting dihydrofolate reductase. Low doses of methotrexate inhibit several other folate-dependent enzymes associated with inflammation regulation to achieve its therapeutic effect in IBD. Besides, methotrexate exhibits an excellent anti-inflammatory effect in the treatment of IBD, inhibiting the production of IL-1, IL-2, IL-6, and IL-8 [74]. Although methotrexate has an excellent clinical effect for the treatment of CD, its treatment of UC patients has not achieved the expected effect. Meanwhile, side effects such as nausea, vomiting, fatigue, diarrhea, leukopenia, hepatic fibrosis, allergic pneumonia, and teratogenicity have been observed with methotrexate. Methotrexate has also been found to have some nephrotoxicity when applied clinically, so proper dose adjustment is needed for patients with renal insufficiency or decreased renal function [75].

### 3.2. Biological Agents

#### 3.2.1. Tumor Necrosis Factor Inhibitors

The tumor necrosis factor-α is a vital cytokine mediating intestinal inflammation, and both TNF-α and its receptors play an essential role in the chronic inflammatory process of IBD. Blocking the action pathway of TNF-α may be a pivotal link to alleviating intestinal inflammation. Currently, anti-TNF-α monoclonal antibodies, which have been approved by the Food and Drug Administration (FDA) for the treatment of IBD, such as infliximab (IFX), adalimumab (ADA), golimumab, and certolizumab pegol (CZP) [76]. Clinical evidence showed that these anti-TNF therapies play a role in restoration of immune homeostasis by affecting the effects of TNF on cytokines, chemokines, acute phase reactants, apoptosis, and inflammation [77]. Despite the effectiveness of anti-TNF in patients with IBD, more than one-third of patients present primary resistance, and Immunogenicity is the main factor affecting the pharmacokinetics and efficacy of anti TNF, antidrug antibodies (ADAs) accelerate anti-TNF monoclonal Abs clearance and shorten their half-life [78]. On the other hand, anti-TNF agents may increase the risk of certain classes of infections and malignancies, yet manageable [79].

#### 3.2.2. Integrin Receptor Antagonists

The inflammatory process in IBD is characterized by extensive infiltration of inflammatory cells in the lamina propria of the intestine, and the migration of inflammatory cells from the vasculature to the site of inflammation must be accomplished by signal communication between α4 integrins on their cell surfaces and adhesion molecules on the vascular endothelium. Therefore, antagonizing integrin receptors on the surface of inflammatory cells is a viable strategy to prevent leukocyte migration from the vasculature to sites of intestinal inflammation. Natalizumab is a chimeric recombinant human IgG4 antibody that exerts its effects on inducing and maintaining clinical remission of CD through anti-α4 integrins interacts with VCAM-1 [80]. However, it has been reported that natalizumab can affect cerebral antiviral immunity, and under certain conditions, can cause progressive multifocal leukoencephalopathy (PML) due to the reactivation of the John Cunningham (JC) virus, thus it was withdrawn from the market in 2005. Subsequently, an α4β7 integrin receptor antagonist that does not activate the JC virus, vedolizumab, came into existence [81].

#### 3.2.3. Interleukin Antagonists

The p40 subunit, shared by IL-12 and IL-23, plays an essential role in the pathogenesis of inflammatory diseases [86]. Ustekinumab, a fully human Ig G1 monoclonal antibody targeting IL-12 and IL-23, can antagonize its action by binding to the p40 subunit shared by both and preventing its binding to the IL-12 Rβ1 receptor on the cell surface, and it reduces T-cell activation [82]. In addition, because it can inhibits pro-inflammatory cytokines, and it reduces T-cell activation, it was more effective than placebo for inducing and maintaining remission in patients with moderate-to-severe ulcerative colitis [83].

Therapeutic options such as combination and sequential use of biologics with conventional drugs or between biologics have also been investigated, and new drugs and new therapeutic options have led to a redefinition of the therapeutic goals of IBD. Although IBD patients currently tolerate biologics well, there are some serious adverse effects, and the safety of long-term use needs further study. Therefore, patient benefits and risk ratios should be weighed and clearly explained to patients when selecting treatment options.

## 4. Stem Cell Therapy for IBD

### 4.1. HSCT and IBD

Hematopoietic stem cells can be isolated from common peripheral blood, bone marrow, and cord blood that migrate directly into damaged tissues or differentiate into epithelial and immunomodulatory cells to restore normal mucosal tissues [87]. The multi-step procedure for HSC treatment starts with a pre-transplant screening, including history and examination, blood tests, serology, colonoscopy, small bowel MRI or pelvic and rectal MRI (in patients with perianal disease involvement), and bone marrow aspiration [88]. The transplantation operation can be performed next by first mobilizing stem cells from the patient or human leukocyte antigen (HLA)-matched donor and stimulating the bone marrow to produce stem cells after lymphocyte removal by infusion of cyclophosphamide. Immediately following leukocyte clearance, CD34^+^ cells are collected from peripheral blood or bone marrow to reach a target number of (3~8) × 10^6^. Finally, transplantation and reconstitution of the immune system were performed. The initial reason for patients with IBD to receive HSCT was the combined hematologic indications (e.g., leukemia and non-Hodgkin’s lymphoma), but improvement in intestinal lesions were also found during transplantation, so HSCT was progressively used in clinical studies for the treatment of IBD [89]. Although the results of the studies were variable, HSCT showed that some patients with UC and CD achieved clinical and endoscopic improvement. With complete immunoablation and hematopoietic stem cell reconstitution, so the body can produce naive lymphoid and myeloid cells, thus reducing T-cell activity against mucosal autoantigens and inflammation.

A study by Burt et al. included 24 patients with CD, with remission rates of 91%, 57%, and 19% at years 1, 3, and 5 [90]. The approach to HSCT in patients with IBD is variable, and the therapeutic approach is evolving. The CD34^+^ peripheral blood stem cells are usually chosen for transplantation in most cases. The CD34 is a hematopoietic progenitor antigen that acts as a cell-cell adhesion factor. Clerici’s study analyzed CD patients with one year of follow-up, most of whom showed clinical and endoscopic complete remission and were maintained for one year without further treatment. The main problem with the previous study was the small sample size and the need for long-term follow-up and randomized controlled studies on a more significant number of patients [91]. Some of the subsequent studies have optimized autologous peripheral blood HSCT. Studies have shown that low doses of cyclophosphamide and granulocyte-stimulating factor improve peripheral blood HSCT outcomes [8].

The discontinuation of all immunosuppressive drugs and clinical remission was found in 11 of 12 patients at 18.5 months of follow-up. It is noteworthy that this study evaluated clinical outcomes but not endoscopic remission. Another study used a similar protocol and looked at the prognosis of 12 patients with refractory CD who received immune pretreatment with autologous HSCT. Stem cell mobilization using cyclophosphamide and granulocyte-stimulating factor revealed that approximately 56% of patients showed clinical and endoscopic improvement after transplantation, and five patients remained in remission at 6-month follow-up [92]. One patient with extraintestinal manifestations of gangrenous sepsis was in complete remission, and another patient had complete fistula closure. However, at a 3-year follow-up, 7 of 9 patients developed recurrent symptoms. The ASTIC study was a multicenter, randomized phase III interventional study initiated by the European Organization for Blood and Marrow Transplantation to evaluate the potential clinical effects of high-dose immunosuppression early and 59-week autologous hematopoietic stem cells. All 45 patients received stem cell mobilization prior to randomization grouping. Only 2 of 23 patients in the stem cell treatment group achieved complete clinical and endoscopic remission, compared with 1 of 22 patients in the control group, showing no statistically significant difference in sustained disease remission between the stem cell treatment and control groups. The trial also noted that all patients experienced non-serious complications, with infection being the most common [93]. One patient in the stem cell treatment group died of sinus obstruction syndrome 20 d after the pretreatment period. Notably, at this stage in clinical studies, autologous HSCT using a non-myeloablative regimen of cyclophosphamide and anti-thymocyte globulin without sorting CD34^+^ cell is the most common transplant pretreatment regimen. Although a limited number of phase I-III HSCT trials have shown promising results, and some patients can benefit in some dimensions, but relapse makes it challenging to classify HSCT as an effective treatment, and larger samples and longer-term efficacy observations are needed [94].

#### 4.1.1. Mechanism and Application of HSCT

Inflammatory bowel disease is an autoimmune disease in which patients can develop impaired immune tolerance in the intestinal tract. Theoretically, HSCT can establish a new immune system in the intestine of IBD patients to avoid autoimmune attacks [95,96]. Inflammatory bowel disease is a polygenic disease with susceptibility sites located on chromosomes 3, 7, 12, and 16. Among them, NOD2/CARD15 is the first susceptibility gene of IBD, which encodes a protein expressed only in peripheral monocytes and can mediate apoptosis as well as induce NF-κB activation. The NOD2/CARD15 mutation can synthesize a large amount of protein in patients and reduce the immune tolerance of patients’ intestines.

The NOD2/CARD15 gene expression may be within hematopoietic stem cells so that allogeneic HSCT can eliminate the genetic defect to reconstitute the immune system [97]. Transplantation of bone marrow stem cells into IBD model mice restores microcirculation in damaged intestinal mucosal tissues and accelerates repair, thus replacing the intestinal epithelium with a regenerative capacity [98]. Primary chemotherapy for HSCT destroys the immune system, including abnormal T-lymphocytes, and returns the immune system to an infantile state. Reselecting T-lymphocytes then regenerate a population of immune tolerant T-lymphocytes in the thymus. The current study observed this T-lymphocyte renewal process after HSCT in multiple sclerosis and systemic lupus erythematosus [99,100]. Clerici et al. found that Treg, TLR4, TNF-α, and IL-10 were higher than expected in CD patients before transplantation, and all of the above cells and cytokines showed a significant decrease after transplantation, suggesting that hematopoietic stem cells have immunomodulatory effects and can reduce inflammatory responses in patients [91]. In contrast, Alexander et al. suggested that the immunomodulatory function of HSCs is related to alterations in the patient’s thymus, i.e., the recipient’s thymus function is reactivated after HSCT and produces primitive immune-tolerant T-lymphocytes, thereby attenuating immune-mediated organismal damage [100].

There are both similarities and differences in the mechanisms of autologous HSCT and allogeneic HSCT (Table 2). Autologous HSCT has been shown to profoundly impact the immune system, including the innate immune system and adaptive immune system. It mainly involves the removal of reactive T-lymphocytes from the patient’s body by chemotherapy and replacing them with more immune-tolerant lymphocytes to rebuild the immune system. Because the grafts are autologous cells and are mostly treated non-cleavage, it is not possible to change the patient’s genetic susceptibility at the genetic level. The IBD can still relapse under specific triggers. Allogeneic HSCT is a complete replacement of the recipient’s immune cells with the donor’s immune cells, thus altering the patient’s susceptibility to IBD at the genetic level, but it is rarely used to treat non-malignant diseases due to its high lethality.

It should be noted that although the future of homozygous HSCT for IBD is worrisome, there is still a group of people for whom this therapy is suitable: namely, low-grade IBD patients with IL-10 gene defects. Kotlarz et al. performed genetic testing for IL-10, IL-10R1, and IL-10R2 in 66 patients with IBD with an age of onset younger than five years and found a total of 16 patients with these genetic defects, 5 of whom underwent allogeneic HSCT and did not experience a relapse during the subsequent 2-year follow-up period [104]. Therefore, for patients with IBD at an early age of onset, IL-10 gene testing can be performed, and if such genetic defects are present, allogeneic HSCT is appropriate.

#### 4.1.2. Safety of HSCT

Although the effectiveness of HSCT has been confirmed in many ways, safety is still an issue that needs to be overcome (Table 2). Autologous HSCT does not fundamentally address the genetic defect but may provide long-term remission due to the absence of triggers after rebuilding the immune system. In contrast, allogeneic HSCT, while potentially curing IBD completely, dramatically increases the set of risks of graft rejection. The immunogenicity of stem cells, the safety of cell culture, the possibility of allogeneic tissue formation, and the transformation of cells during in vitro expansion are essential aspects that can affect the success and safety of transplantation. Snowden et al. reviewed clinical trials of hematopoietic stem cell transplantation for immune-mediated diseases included in the 1997–2009 database and found that the one-year and five-year survival rates for autologous HSCT were 85% and 78%, respectively, while the one-year and five-year survival rates for allogeneic HSCT were 87% and 65% [96]. Here, infection is the most important cause of death in patients. The above data show that the safety of HSCT in the treatment of immune-mediated diseases is still a pressing issue.

### 4.2. Mesenchymal Stem Cells and IBD

It is well known that MSCs, also referred to as adult stem cells are multipotent adult stem cells derived from the ectoderm and mesoderm. Mesenchymal stem cells are present in the whole body tissues, mainly in connective tissue and organ mesenchyme [105]. Most of these cells express CD105, CD73, and CD90 and can differentiate into osteoblasts, chondroblasts, and adipocytes in vitro [106]. Similar to hematopoietic stem cells, MSCs also have proliferative and differentiation, immunomodulatory and nutritional roles and occupy an essential position in regenerative medicine. The use of MSCs has been reported for the treatment of acute hormone-resistant graft-versus-host disease, osteogenesis imperfecta in children, ankylosing spondylitis, disc herniation, and some neurological disorders [107].

Mesenchymal stem cells induce cell cycle arrest and apoptosis in lymphocytes mainly by releasing various soluble factors (such as cytokines, chemokines, and growth factors) to achieve immune regulation and inflammation control, leading to tissue regeneration [108]. Bone marrow mesenchymal stem cells (BMMSCs) and adipose mesenchymal stem cells (ASCs) were found to enhance the activity of immunosuppressive molecules, such as hepatocyte growth factor (HGF), TGF-β, prostaglandin E2 (PGE-2), indoleamine 2,3-dioxygenase (IDO), IL-6, IL-10, nitric oxide, heme oxygenase-1, HLA-G5 [109]. It is worth noting that the immunomodulatory effect of MSC is more prominent in the inflammatory environment, as reflected by the presence of certain pro-inflammatory factors (such as TNF-α and IFN-γ) that are prerequisites for the immunomodulatory effect of stem cells [110].

The clinical trials related to MSCs for IBD have also made some progress. The BMMSCs are the most well-researched MSCs, but the extraction of BMMSCs is harrowing due to their low content. Moreover, the number of BMMSCs gradually decreases with age, and its clinical application has some limitations. Since then, with the advent of ASCs, the defects of BMMSCs have been overcome and have good application prospects. Mesenchymal stem cells for intestinal luminal IBD has been reported less frequently, while Liang et al. retrospectively analyzed seven patients with refractory IBD (3 with UC and 4 with CD) transplanted with allogeneic BMMSCs showed symptom reduction and reduced inflammatory activity [111]. In addition, fistulas are an essential complication of CD, affecting more than 50% of patients. The majority of fistulas in CD patients occur in the perianal region, and 21% of Crohn’s disease patients develop anal fistulas ten years after diagnosis. Local injection of MSCs can regulate the epithelial-mesenchymal transition of fistula cells by secreting growth factors that promote wound healing. Mesenchymal stem cells can be mixed with fibrin glue and injected directly into the fistula locally, or they can be attached to a biological scaffold and inserted into the fistula [112]. For the treatment of fistulas due to IBD, García-Olmo et al. reported in 2003 a case of ASCs successfully treating a rectovaginal fistula caused by CD, suggesting that stem cell therapy could potentially be applied to such conditions [113]. This team subsequently reported a phase II clinical trial in 2009, which found significantly higher fistula closure rates in the ASCs combined with fibrin than in the fibrin alone group by comparing local injections of ASCs combined with fibrin to the treatment of complex fistulas [114]. However, after up to 3 years of follow-up, only a tiny proportion (7/24) of the combined treatment group was recurrence-free, so the long-term effect of ASCs remains uncertain [115]. Based on these studies, the team reported a phase III clinical trial of ASCs for perianal fistulae in 2012. They compared the efficacy of ASCs, fibrin, and ASCs combined with fibrin and found that the fistula closure rates in the three groups were 39.1%, 43.3%, and 37.3%, respectively, after 24~26 weeks (*p* = 0.79).

In contrast, the fistula closure rates in the three groups after one year were 57.1%, 52.4%, and 37.3% (*p* = 0.13), with no severe side effects in any of the three treatments [116]. Since the first report of successful rectovaginal fistula healing after MSC injection in 2003, several clinical trials have demonstrated the safety and efficacy of this emerging therapy for fistula-type CD. Trials to date have used both allogeneic and autologous bone marrow MSCs, administered at different doses, given as single or repeated injections, and delivered using a scaffold, all of which have implications for safety and efficacy.

#### 4.2.1. Mechanism of MSCs for IBD

Mesenchymal stem cell treatment of IBD has been studied in many preclinical studies trials in animal models of IBD. When MSCs are injected intravenously, they can reach the site of intestinal injury and colonize the intestinal mucosa to control the development of inflammation locally, improve local microcirculation, and repair damaged tissues [117]. Studies have shown that MSC treatment of IBD is mainly carried out through local microcirculation construction, colonization and repair, and immunomodulation (Table 3).

##### Local Microcirculation Construction

In IBD, chronic inflammation of the colon is affected by microvascular dysfunction and endothelial barrier damage, resulting in sustained inadequate perfusion of colonic tissue and poor healing of colonic tissue. Mesenchymal stem cells play a role in tissue repair mainly by differentiating into vascular endothelial cells (ECs), promoting angiogenesis, and improving local blood supply. It has been shown that MSCs induced containing vascular endothelial growth factor (VEGF), primary fibroblast growth factor (FGF), insulin-like growth factor (IGF), epidermal growth factor (EGF), ascorbic acid, and heparin can effectively differentiate into EC and form capillary networks in vitro [118]. Mesenchymal stem cells may induce EC proliferation by producing multiple pro-angiogenic factors that promote neointimal formation [119]. Britten’s experiments showed that BMMSCs could colonize and differentiate into intestinal subepithelial myofibroblasts in radiation enteritis intestinalis, which can directly secrete and paracrine a variety of cytokines that make up the intestinal stem cell microenvironment [120]. These cytokines can be involved in processes that regulate the proliferation of intestinal ECs, differentiation of ECs, protection of mucosa, promotion of wound healing, assistance in water and electrolyte transport, promotion of extracellular matrix metabolism, and growth of basement membranes [134]. In addition, it has also been demonstrated that BMMSCs can differentiate into vascular endothelial-like cells under certain conditions.

##### Fixation Repair

The intestinal mucosal epithelial barrier consists of a mucus layer, intestinal epithelial cells, TJs, and the submucosal lamina propria. It can prevent the invasion of pathogens while allowing the selective passage of nutrients. Immunologically active cells in the epithelium can respond to many signals from the organism, including those from intestinal microbes and the immune system. Abnormal intestinal mucosal barrier function at the onset of IBD leads to increased mucosal permeability and disruption of the intestinal mucosal barrier. Intestinal stem cells (ISCs) are known to have the ability to significantly promote efficient renewal and repair of the intestinal epithelium [20]. Mesenchymal stem cells not only stimulates the proliferation of intestinal epithelial cells but also increase the number of Lgr5^+^ ISCs [121] by a mechanism that may be related to increased activation of the Wnt/β-catenin transduction pathway [135]. In addition, MSCs injected in the tail vein nest to damaged sites of the intestine in mice with experimental colitis and repair the integrity of the intestinal epithelial barrier through recombinant cellular [122]. Mesenchymal stem cells secrete factors that accelerate airway epithelial repair by stimulating the endogenous repair and regenerative capacity of lung cells [136]. Therefore, the ability of MSCs to stimulate the proliferation of intestinal stem cells, promote endogenous repair of IBD intestinal epithelial cells and inhibit apoptosis may be closely related to their secreted cytokines or chemokines [123].

After transplantation of BMMSCs in UC rats, the migration rate of BMMSCs increased with an increasing injury during the repair injury period and decreased significantly during the recovery period [137]. It was also confirmed that transplanted BMMSCs could colonize the intestine of rat models and participate in the repair process [122]. Mesenchymal stem cells can grow locally and differentiate into cells with an epithelial cell phenotype, while relevant in vivo experiments need further refinement.

##### Immunomodulation

Inflammatory bowel disease is a chronic inflammatory disease associated with autoimmunity, and abnormalities in the immune system of the intestinal mucosa are a crucial factor in its development. When tissues are damaged, MSCs play an essential role in tissue repair by modulating the immune response to improve the inflammatory microenvironment.

Macrophages

The intestine contains a massive reservoir of macrophages that are essential for maintaining mucosal homeostasis and epithelial renewal, as well as being an essential component of protective immunity and involved in the pathological processes of IBD [138]. It was shown that TNF-α-stimulated gene-6 (TSG-6) secreted by human adipose-derived MSCs (ADMSCs) could improve the DSS-induced experimental colitis model by inducing a shift in macrophage phenotype from M1 to M2 to regulate inflammatory cytokine expression [124]. Subsequently, this team demonstrated that canine ADMSCs exerted a similar effect [139]. Exosomes from human umbilical cord MSCs (UCMSCs) inhibited IL-7 expression in peritoneal macrophages, suggesting that exosome treatment may affect IL-7 expression in macrophages and thus attenuate DSS-induced colitis in mice [140]. An in vitro experiment showed that human MSCs could produce prostaglandin E2 (PGE2) in vitro, thereby altering the macrophage phenotype, but further studies are needed to confirm whether this has an effect on macrophages in inflammatory bowel disease [141].

Dendritic Cells

Dendritic cells in the colon display a tolerogenic phenotype under a steady state. Activated DCs produce large amounts of pro-inflammatory cytokines and express high co-stimulatory molecules (e.g., CD40, CD80, and CD86), promoting intestinal inflammation [142]. It has been shown that peripheral blood plasma cell-like DCs from IBD patients exhibit an activated and mature phenotype. In contrast, peripheral blood plasma cell-like DCs from mesenteric lymph nodes and controls have lower activity [143], suggesting the maturation and activation of DC play an important role in IBD progression. Mesenchymal stem cells can inhibit DC maturation and alter their secretory profile in response to tolerogenic phenotypes, leading to reduced production of pro-inflammatory factors and increased anti-inflammatory factors, resulting in diminished T-cell activation [125]. However, the detailed mechanism of action has not been fully elucidated. Recent studies have shown that β-galactoside binding lectin-1 secreted by bone marrow MSCs upregulates Gal-1 expression in DCs and stimulates the development of a tolerogenic immune phenotype in DCs, with the potential mechanism being regulation of the MAPK signaling pathway in DCs, thereby inhibiting DC function [126].

T-cell subsets

When the balance of the body’s immune system is disrupted during an IBD episode, Th-cells can be activated by intestinal secretions, intestinal bacteria, and immune-related proteins, and Th1 cells cause mucosal inflammation. In contrast, Th2 cells can suppress excessive inflammatory responses by regulating the ratio of Th1 to Th2 [39]. Mesenchymal stem cells exert immunomodulatory effects by suppressing Th1 type pro-inflammatory factors and enhancing the expression of Th2 type anti-inflammatory factors [127]. In addition, MSCs can induce differentiation and maturation of Th2 cells by expressing indoleamine 2,3-dioxygenase (IDO), leading to tryptophan depletion and the production of tryptophan metabolites, promoting apoptosis of Th1 cells [128].

The CD4^+^ T-cell subsets, which are pathologically underlying the development and progression of IBD, abundant in the lamina propria, can be activated upon contact with food and microbial antigens in the gastrointestinal tract. The application of MSC in an animal model of IBD resulted in a significant decrease in IL-17 expression, a decrease in IL-17/IFN-γ-producing cells in the spleen, and a decrease in the accumulation of neutrophils in the intestine [129]. In addition, MSC can inhibit the STAT3 pathway and decrease the expression of IL-17, thus reducing the differentiation of Th17 cells and promoting the repair of IBD-injured tissues [130].

The regulation of intestinal inflammation is closely related to the presence and function of Treg cells, which can suppress excessive inflammatory responses, secrete anti-inflammatory factors IL-10 and TGF-β, and express fork-like transcription factor P3 (FOXP3) [144]. High FOXP3 expression can directly or indirectly inhibit effector T-cells through activated antigen-presenting cells, leading to their immune tolerance to autoantigens and protecting tissues from injury [145]. Mesenchymal stem cells promote the differentiation of Tregs cells and inhibits the differentiation of Th1 and Th17 cells in CD4^+^ T-cells mainly through direct contact with cells or by producing anti-inflammatory factors while inhibiting the secretion of pro-inflammatory factors [131]. Therefore, the improvement of colitis after MSC application for IBD may be related to the direct or indirect regulation of the inflammatory response by Treg cells.

The balance between Th17 cells and Tregs cells is vital. One of the essential mechanisms by which MSCs improve the condition of IBD patients is the inhibition of Th17 cell-mediated immune response and enhancement of Tregs cell function in vivo [132]. Th17 cells exert pro-inflammatory effects by expressing retinoic acid-related orphan receptor γt (RORγt) and secreting factors such as IL-17 and IL-6. In contrast, Treg cells play a role in the anti-inflammatory process by expressing FOXP3 and secreting factors such as IL-10 and TGF-β [133]. The Th17 cells and Tregs cells can switch and regulate each other during cytokine-dependent differentiation. Studies have shown that the phosphorylation levels of STAT3 and STAT5 are closely related to the balance between Th17 cells and Tregs cells. After applying MSC treatment, STAT3 phosphorylation is inhibited while STAT5 phosphorylation is induced. The STAT3 phosphorylation increases RORγt expression, leading to increased pro-inflammatory cytokine expression levels. The STAT5 phosphorylation inhibits Th17 cell differentiation and cytokine IL-17 production while increasing Tregs cell differentiation and FoxP3 expression [130]. Therefore, by adjusting the STAT3/STAT5 signaling pathway, the balance between Th17/Tregs in IBD patients can be regulated.

MSCs-derived exosomes

Mesenchymal stem cells can exert immunomodulatory effects in a paracrine manner, and one of the essential paracrine substances is exosomes. Exosomes are nanoscale vesicles with a phospholipid bilayer structure secreted by cells, which contain proteins, microRNA, mRNA, and other substances [146]. It has been shown that MSC-Exo can polarize macrophages to the M2 type, thereby downregulating the inflammatory response and maintaining the integrity of the intestinal mucosal barrier [147]. In addition, MSC-Exo can inhibit the proliferation and differentiation of T-cells and promote apoptosis of activated T-cells. At the same time, it can also aggregate Tregs cells and eventually lead to decreased expression of pro-inflammatory cytokines such as TNF-α, and IFN-γ and increased expression of anti-inflammatory cytokines such as IL-10, and TGF-β, thus promoting the repair of damaged intestinal tissues [148]. In addition, MSC-Exo can inhibit oxidative stress by decreasing the expression of oxidation-inducing factors such as MPO and MDA and increasing the expression of antioxidant factors such as GSH and SOD, thus alleviating the symptoms of IBD patients [147].

#### 4.2.2. Safety of Mesenchymal Stem Cells

The safety issue of MSCs in the treatment of IBD is still not conclusive. Based on the proliferative and multidirectional differentiation potential of MSCs, it has been suggested that such therapies can increase the risk of tumorigenesis. There are also related animal experiments in which ASCs were infused into mice simultaneously with tumor cells, and it was found that the tumors appeared earlier and were more prominent in mice [149]. However, MSCs have also been found to inhibit tumorigenesis. Nasuno et al. found that MSCs achieve tumor suppression through mechanisms such as induction of apoptosis and control of cell division [150] Chen et al. found that BMMSCs reduce the risk of tumor formation due to colitis and that MSCs reduce inflammation at the bulk level, decrease the expression of pro-inflammatory factors, downregulate STAT3 phosphorylation expression, reduce tumor number, and decrease tumor load [151]. In addition to carcinogenicity, other adverse effects of MSCs have been systematically evaluated. Lalu et al. performed a systematic review of prospective clinical trials on the systematic use of MSCs (arterial and intravenous injection) until 2011 and found that, except for the transient fever, the use of MSCs did not show significant side effects and was a relatively safe treatment [152].

### 4.3. Embryonic Stem Cells and IBD

Embryonic stem cells (ESCs) are an important research topic. Embryonic stem cells are pluripotent cells obtained from embryos, which have the potential of self-renewal and can differentiate into intestinal epithelial cells and immune cells [153]. In addition, embryonic stem cells also have immunomodulatory and inflammation-relieving properties, have a faster growth rate and more extraordinary differentiation ability than adult stem cells, and may become another type of stem cell used for the treatment of IBD after adult stem cells [154]. But there is still a lack of literature on ESCs for the treatment of IBD. When studying the effect of mouse embryonic stem cell transplantation on the severity of colitis and immune imbalance in pyrrolidone induced colitis in IL-10^−/−^ KO mice, Srivastava et al. proved that induced pre-differentiated ESCs were able to colonize the small intestine, colon, and liver, as well as alleviate inflammation through mechanisms such as repairing damaged epithelium and reconstituting the immune system [155]. The biggest challenge in the current application of ESCs is the control of differentiation, i.e., the induction of cells to grow and differentiate along the desired pathway by a specific external environment.

## 5. Conclusions

Inflammatory bowel disease is a chronic inflammatory disease, which seriously affects people’s daily life and health, further insights in the pathogenesis of IBD are required. At present, the therapeutic drugs used in clinical practice mainly include small molecule medications and biological agents. However, some of these drugs have significant side effects. The goal of an effective treatment in IBD is to arrest the disease progression aiming to achieve complete remission of IBD.

Stem cell therapy for IBD has a good application prospect. Stem cell research has made tremendous advancements over the past decades (Table 4). A certain number of clinical trials have confirmed the efficacy of HSC and MSCs in IBD, leading to a cure in some patients. However, the existing treatments need to be improved, continued in-depth research is required to fully understand the complex cellular mechanisms. The HSCT requires rigorous patient screening and must be performed at technically mature institutions with the ability to manage the primary disease and the complications of transplantation. The success of MSC transplantation depends on several factors, the selection and optimization of which need to be further investigated. Therefore, stem cell therapy for IBD is still in its infancy, and some of the findings are inconclusive. Thus, many issues need to be investigated in-depth, including the mechanisms of stem cells, the safety, and long-term efficacy of transplantation for colitis, to maximize therapeutic potential during the treatment of inflammation induced intestinal disorders. Besides, issues regarding the isolation and purification techniques, and immune rejection of stem cells still need further research [156]. Partial results from the ongoing Phase 3 clinical trial will provide a valuable roadmap and standardized process for future stem cell therapy. Cell therapy should not be limited to hematopoietic stem cells and mesenchymal stem cells, and other cell therapy methods continue to be explored preclinically. Future research needs to focus on improving the safety and feasibility demonstration, and with the goal of improving the quality of life of patients. In conclusion, the wider usage of stem cells in the treatment of IBD are safe and highly effective in the long term.

## Figures and Tables

**Figure 1 ijms-23-08494-f001:**
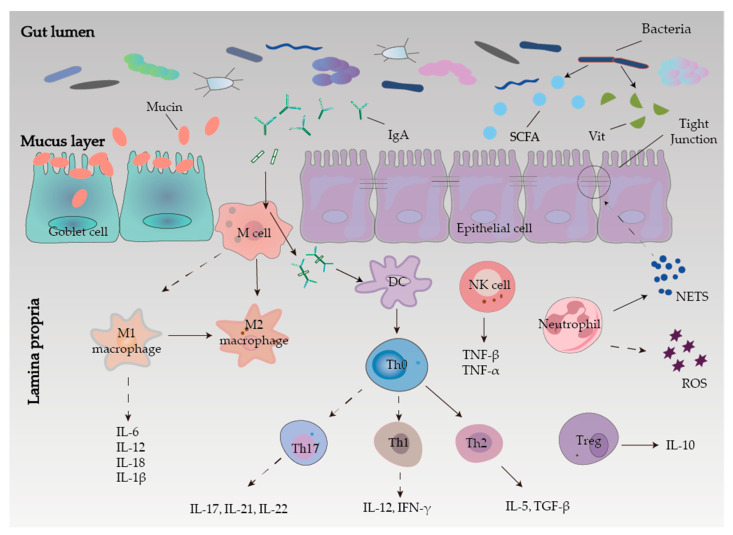
Intestinal mucosal barrier function.

**Figure 2 ijms-23-08494-f002:**
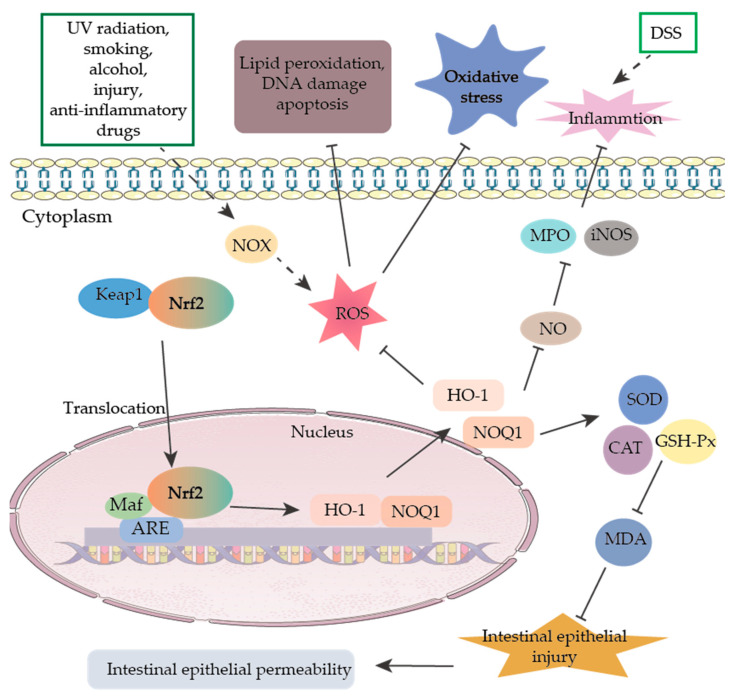
Regulation of oxidative stress through Nrf2 signaling pathway.

**Figure 3 ijms-23-08494-f003:**
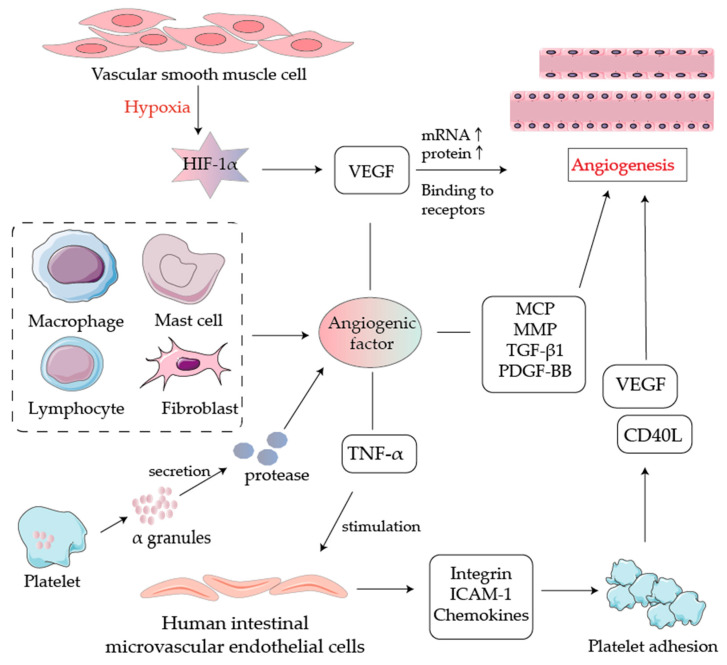
Regulation of angiogenesis.

**Table 1 ijms-23-08494-t001:** Summary of novel medications intended for the management of IBD.

Treatments	Molecule Group	Compound	Mechanisms of Action	Limitations	References
Small molecule medications	Amino Salicylate	5-ASA	Antibacterial action Inhibits prostaglandin and NF-κB activity Scavenge reactive oxygen metabolites	Adverse gastrointestinal events: nausea, and vomiting	[70]
	Glucocorticoids	Prednisone Hydrocortisone	Interact with glucocorticoid receptors in the nucleus	Osteoporosis, systemic edema, infections, and a high drug dependence	[71,72,73]
			Inhibit the expression of adhesion molecules		
			Prevent the transport of inflammatory cells to the intestine		
	Immunomodulators	Methotrexate	Inhibit several enzymes involved in the folate metabolic pathway	Nausea, vomiting, fatigue, diarrhea, leucopenia, liver fibrosis, allergic pneumonia, teratogenicity, and certain nephrotoxicity	[74,75]
			Play an anti-inflammatory role by inhibiting the production of IL-1, IL-2, IL-6, and IL-8		
Biological agents	Tumor necrosis factor inhibitors	Infliximab Adalimumab Golimumab Certolizumab pegol	Affect the downstream effects of TNF on cytokines, chemokines, acute phase reactants, apoptosis, and inflammation	Some patients present primary resistance, and may increase the risk of certain uncommon classes of infections and malignancies	[76,77,78,79]
	Integrin receptor antagonists	Natalizumab	Induce and maintaining clinical remission of CD through anti-α4 integrins interacts with VCAM-1	Affect cerebral antiviral immunity, and can cause a fatal brain infection due to the reactivation of the JC virus	[80,81]
	Interleukin antagonists	Ustekinumab	Bind to the p40 subunit shared by both and preventing its binding to the IL-12β1 receptor on the cell surface	Potential risks are not known	[82,83]
			Inhibit pro-inflammatory cytokines		
			Reduce T-cell activation		

**Table 2 ijms-23-08494-t002:** Different types, mechanisms, advantages, and disadvantages of HSCT for IBD.

Types	Mechanism	Advantage	Deficiency	References
Autologous HSCT	The removal of reactive T-lymphocytes by chemotherapy and replacing them with more immune-tolerant lymphocytes to rebuild the immune system	Provide long-term remission	It is not possible to change the patient’s genetic susceptibility at the genetic level	[101,102]
Allogeneic HSCT	A complete replacement of the recipient’s immune cells with the donor’s immune cells	Alter the patient’s susceptibility to IBD at the genetic level	High lethality, and increases the set of risks of graft rejection	[101,103]

**Table 3 ijms-23-08494-t003:** The mechanism of action of MSC transplantation in the treatment of IBD.

Biological Effects	Mechanism of Action	References
Local microcirculation construction	Differentiation into vascular endothelial cells, and form vessel like structures in vitro	[118]
	Induce EC proliferation by producing multiple angiogenic factors	[119]
	Directly secrete and paracrine a variety of cytokines	[120]
Fixation repair	Activate the Wnt/β-catenin signaling pathway, increase the number of Lgr5+ ISCs	[121]
	Repair the integrity of the intestinal epithelial barrier through recombinant cellular	[122]
	Inhibit apoptosis	[123]
Immunomodulation	Alter the macrophage phenotype from M1 to M2	[124]
	Upregulate Gal-1 expression, inhibit DC maturation, in-crease anti-inflammatory factors, diminish T-cell activation	[125,126]
	Regulate the ratio of Th1 to Th2	[127,128]
	Adjust the balance between Th17/Tregs	[129,130,131,132,133]

**Table 4 ijms-23-08494-t004:** Clinical trials in stem cell-based therapies for Inflammatory Bowel Disease.

Disease	Clinical Indication	Source of Stem Cells	Number of Patients Enrolled	Follow-Up Period	Outcome	Author and Time	Reference
Refractory CD	Have failed treatment with corticosteroids, mesalamine, met- ronidazole, azathioprine (or 6-mercaptopurine), and mono- clonal antibody to TNF receptor (infliximab).	Autologous HSCT	12	7–37 months	11 patients entered a sustained remission. After a median follow-up of 18.5 months, only 1 patient has developed a recurrence of active CD, which occurred 15 months after HSCT	Oyama, Y. et al., 2005	[8]
Severe CD	Refractory to conventional therapies including anti-TNF inhibitor	Autologous nonmyeloablative HSCT	24	1–5 years	Eighteen of 24 patients are 5 or more years after transplantation	Burt, R.K. et al., 2010	[90]
Active moderate-severe CD	Refractory or intolerant to various conventional treatment schedules including corticosteroids and at least 2 immunosuppressors	Autologous HSCT	7	1 year	Most of whom showed clinical and endoscopic complete remission and were maintained for one year without further treatment	Clerici, M. et al., 2011	[91]
Refractory CD	Intolerance or failure of conventional therapies including immunosuppressors and at least one anti-TNF antibody	High-dose immunosuppression and autologous peripheral blood stem cell transplantation (autoPBSCT)	12	0.5–10.3 years	5 patients achieved a clinical and endoscopic remission within 6 months after autoPBSCT. However, relapses occurred in 7/9 patients during follow-up, but disease activity could be controlled by low-dose corticosteroids and conventional immunosuppressive therapy	Hasselblatt, P. et al., 2012	[92]
Refractory CD	Treatment with 3 or more immunosuppressants or biological agents and corticosteroids leads to impaired quality of life and is not suitable for surgery	Autologous HSCT	45	1 year	Compared with conventional therapy, did not result in a statistically significant improvement in sustained disease remission at 1 year and was associated with significant toxicity	Hawkey, C.J. et al., 2015	[93]
Early onset IBD	IL-10 gene defect	Allogeneic HSCT	66	2 years	Allogeneic HSCT was performed in 5 patients to induce sustained clinical remission	Kotlarz, D. et al., 2012	[104]
Refractory IBD	4 CD, 3 UC	Allogeneic MSCT	7	Mean 19 mouths	Diarrhea frequency and abdominal pain/ cramps gradually improved in all the seven patients, accompanied by a significant reduction in CD Activity Index scores in CD patients and Clinical Activity Index scores in UC patients	Liang, J. et al.,	[111]
Luminal CD	With infliximab- or adalimumab-refractory, endoscopically confirmed	Allogeneic MSCT	16	42 days	In a phase 2 study, administration of allogeneic MSCs reduced CDAI and CDEIS scores in patients with luminal CD refractory to biologic therapy	Forbes, G.M. et al., 2013	[112]
CD	Had a recurrent rectovaginal fistula	ASCs	1	3 mouths	Since the surgical procedure 3 month ago the patient has not experienced vaginal flatus or fecal incontinence through her vagina	Garcia-Olmo, D. et al., 2003	[113]
Complex perianal fistulas	Had a complex perianal fistula (either of cryptoglandular origin or associated with CD) with a visible external opening.	Administration of expanded ASCs in combination with fibrin glue	24	1 year	Combination therapy appears to achieve higher rates of healing than fibrin glue alone	Garcia-Olmo, D. et al., 2009	[114]
perianal fistulas	Had received at least one dose of treatment (ASCs plus fibrin glue or fibrin glue alone)	ASCs plus fibrin glue or fibrin glue alone	49	3 years	A low proportion of the stem cell-treated patients with closure after the procedure remained free of recurrence after more than 3 years of follow-up	Guadalajara, H. et al., 2011	[115]
complex fistula-in-ano	Complex fistula-in-ano	A dose of 20 or 60 million ASCs alone or in combination with fibrin glue	200	1 year	Achieving healing rates of approximately 40% at 6 months and of more than 50% at 1-year follow-up	Herreros, M.D. et al., 2012	[116]

## Data Availability

Not applicable.

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
