# Peer review of "Stem Cell-Based Therapies for Inflammatory Bowel Disease"

_ijms, 2022, doi:10.3390/ijms23158494_

Round 1

Reviewer 1 Report

The manuscript submitted by Zhang HM and collaborators entitled ‘Emerging Roles of Stem Cells in Inflammatory Bowel Disease’ is interesting and would be of great interest for the audience of IJMS. The manuscript is well written and would require careful revision of the references cited within the text as in some instances no citation number is found within the ‘(   )’ and in other cases, the message ‘ Error! Reference source not found’. The title of the manuscript is somehow misleading, as the actual roles of stem cells in IBD are not discussed in detail. Instead, ‘Stem cell-based therapies for inflammatory bowel disease’ would be more adequate.  Regarding the content it would be better to significantly reduce the content presented in paragraphs: ‘2. The pathogenesis of IBD’ and ‘3. Pharmacological treatment of IBD’ from page 3 to page 13.

The tables 2 and 3 would need to add an additional column citing the most important references where the key information summarised in each line can be found.

The use of embryonic stem cells are only mentioned very briefly in the Discussion. A fourth paragraph on ‘Embryonic Stem cells in IBD’ where details on their use in preclinical studies would be required.

Minor points to correct are:

1.       Reference 37, is a study related to asthma, the authors should confirm whether the reference was correctly cited.

2.       Reference 87, is a study related MSCs and not HSC, the authors should modify this.

3.       Page 15, line 569, Kotlarz D et al. study not found in the reference list.

4.       Page 19, line 729, replace ‘galactose lectin-1’ by ‘ beta-galactoside binding lectin-1’.

5.       Page 21, line 821, as mentioned above, a specific paragraph dedicated to ‘Embryonic Stem cells’ should be added and deleted in the Discussion.

 The manuscript would also benefit if a final paragraph including major achievements reached by the use of stem cell-based therapies in the clinic and some final remarks highlighting what is coming in this field in the near future.

Author Response

We are very grateful to the Reviewer for reviewing the paper so carefully. We have tried our best to improve the manuscript and have modified some questions. The references in the text were edited with Endnote software. Sometimes this happens when using different software to open. Therefore, we have made some revisions to the references. The title of the manuscript has also been changed as requested by the Reviewer. The content aspect of the article has been reduced. Table 2 and Table 3 have added an extra column to cite references.

Minor points:

1. Reference 37 has been replaced.

2. Reference 87 has been revised.

3. The research article by Kotlarz D et al. has been reflected in the reference list (Reference 101).

4. The description of "beta-galactoside binding lectin-1" has been replaced.

5. We found that there are very few articles on embryonic stem cells and IBD after various related literature. So we modified this part. Since embryonic stem cells are also a type of stem cell, they have a certain effect on the treatment of IBD to a certain extent. Therefore, we did not delete this part but put it at the end of the article. Thanks to the Reviewer for the suggestion.

Finally, based on the Reviewer's suggestion, we revised the conclusion section.

Thanks for the opinion!

Reviewer 2 Report

The authors review the existing data regarding the applications of stem cells therapy in IBD. The theme is interesting and novel in the field of IBD. 

However, the text itself is too long, looking more like a book chapter rather than a review. I do not understand the role of reviewing the other treatment options for IBD, as they are already known?

Moreover, the search methodology used for this review is not described. Also, I would have appreciated a table showing the different studies taken into consideration and their individual results, nr of patient included, type of stem cells used etc. 

There is no conclusion at the end of the paper. 

Please revise accordingly, as the theme chosen is really interesting and the work behind this paper is worth publishing. 

Author Response

We are very grateful to the Reviewer for reviewing the paper so carefully. We have carefully considered the suggestion of the Reviewer and made some changes. 

According to the reviewer's suggestion, we have made some changes and deletions to the article. In our view, review articles facilitate interdisciplinary research and help readers quickly grasp cutting-edge topics. If the readers read this article, they can quickly understand the relevant mechanisms and treatment methods of IBD. In the description of other treatment options for IBD in this article, we can see that the treatment of IBD is not easy, and the need for stem cell therapy can be revealed by analyzing it from various aspects.

The search methods for this review are listed. And stem cell-related tables have also been highlighted in the text.

The conclusions of the article are listed at the end.

Thanks for your opinion!

Round 2

Reviewer 1 Report

The authors have addressed most of the points that arose in the first revision of the manuscript improving its quality. Still, there are some minor points that would need to be addressed in the revised manuscript. These are:

1.       Page 3, line 114, there is a missing reference.

2.       Table 1, would also need to include references in an additional column as in tables 2 and 3.

3.       Headings for table 1 could be: Treatments / Molecule group / Compound / Mechanism of action / Limitations

4.       Page 15, line 596, remove ‘(   )’.

5.       The content in Table 2 corresponds to a single reference (Ref. 100). Such a table may not be needed and just discussed within the main text in a dedicated paragraph.

6.       Page 16, line 617, remove ‘(   )’.

7.       Table 3, replace the title of the table with ‘The mechanism of action of MSC transplantation in the treatment of IBD’.

8.       Headings for table 2 could be: Biological effects / Mechanism of action / References

9.       In table 2, since the mechanism of action for References 116 and 118 are similar consistency in the description of the mechanism of action should similar. In case, there are differences these should be clearly stated.

10.   In Table 2, the mechanism of action for References 122 and 123 could be simplified.

11.   Page 19, the first paragraph from lines 729 to 734 should be moved after the second paragraph once the detailed description of the epithelial barrier is described.  

12.   Page 22, line 890, in Conclusion, replace ‘little is known about its clear pathogenesis’ with ‘further insights in the pathogenesis of IBD are required.’

13.   Page 22, line 894, the last sentence could be rephrased as ‘The goal of an effective treatment in IBD is to arrest the disease progression aiming to achieve complete remission of IBD.’

14.   Page 23, line 917, the last sentence could be rephrased as ‘… are safe and highly effective in the long term.’

15.   Title for Table 4 could be: ‘Clinical trials in stem cell-based therapies for IBD’.

16.   Re-organised Table 4 according to the following headings: Clinical Indication / Source of stem cells / Number of patients enrolled / Follow-up period / Outcome / Authors and publication (Reference).

Author Response

Thank you for your comments concerning our manuscript. Those comments are all valuable and very helpful for revising and improving our paper, as well as the important guiding significance to our research. We have studied the comments carefully and have made corrections which we hope meet with approval. The main corrections in the paper and the responses to the reviewer’s comments are as follows:

1. Reference has been added on page 3, line 114.

2. As with Tables 2 and 3, Table 1 already lists references in additional columns.

3. The title of Table 1 has been changed as requested by the Reviewer.

4. In the PDF version, the "( )" on page 15, line 596 may be due to a display error. In the Word version, "Table 2" is marked within "( )" on page 15, line 596.

5. Contents in the table We have added more references for a better description.

6. In the PDF version, the "( )" on page 16, line 617 may be due to a display error. In the Word version, "Table 2" is marked within "( )" on page 16, line 617.

7. The table titles have been changed as requested by the Reviewer.

8. The table's title has been changed to the Reviewer's content.

9. We have refurbished the table due to similar mechanisms of action described in ref.

10. The mechanism of action of references 122 and 123 have been simplified.

11. As requested by the Reviewer, page 19, the first paragraph of lines 729 to 734 has been moved.

12. Line 890 on page 22 has been modified.

13. The last sentence on page 22, line 894 has been revised.

14. The last sentence on page 23, line 917 has been revised.

15. The title of the table has been revised.

16. The title of the table has been modified as requested by the Reviewer.

Special thanks to you for your good comments.

Reviewer 2 Report

Thank you for your efforts to comply with my recommendations!

Accept in present form

Congratulations!

Author Response

Thank you very much for your evaluation of this article. May the joy and happiness around you today and always.